# Consumer Perception of Food Quality and Safety in Western Balkan Countries: Evidence from Albania and Kosovo

**DOI:** 10.3390/foods10010160

**Published:** 2021-01-14

**Authors:** Rainer Haas, Drini Imami, Iliriana Miftari, Prespa Ymeri, Klaus Grunert, Oliver Meixner

**Affiliations:** 1Department of Economics and Social Sciences, Institute of Marketing & Innovation, University of Natural Resources and Life Sciences, 1180 Vienna, Austria; rainer.haas@boku.ac.at; 2Faculty of Economics and Agribusiness, Agricultural University of Tirana, 1025 Tirana, Albania; dimami@ubt.edu.al; 3Faculty of Tropical Agri Sciences, Czech University of Life Sciences Prague and CERGE EI, 16500 Prague, Czech Republic; 4Department of Agricultural Economics, Faculty of Agriculture and Veterinary, University of Prishtina, 10000 Prishtina, Kosovo; Iliriana.Miftari@uni-pr.edu (I.M.); prespaymerii@gmail.com (P.Y.); 5Centre for Research on Customer Relations in the Food Sector MAPP, Aarhus University, 8210 Aarhus, Denmark; klg@mgmt.au.dk

**Keywords:** food safety, food quality, Kosovar consumers, Albanian consumers, Western Balkan countries, bootstrapping

## Abstract

Domestic food markets are of significant importance to Kosovar and Albanian companies because access to export markets is under-developed, partly as a result of the gaps in food safety and quality standards. Kosovar and Albanian consumers’ use of food safety attributes and their evaluation of the quality of domestic food versus imported food are the research objectives of this study. The paper is based on a structured consumer survey of 300 Kosovars and 349 Albanians analyzing their perceptions of issues related to food safety and quality, measured through two respective batteries of items using a 5-point Likert scale. We used the *t*-test to identify differences between populations, correlation analysis and the bootstrapping method. Despite the prevalent problems with food safety, consumers in both countries consider domestic food to be safer as well as of higher quality than imported products. Kosovars are more likely than Albanians to perceive domestic food products to be significantly better than imported products. Female and better educated consumers use information related to food safety more often. Expiry date, domestic and local origin, and brand reputation are the most frequently used safety and quality cues for both samples. International food standards such as ISO or HACCP are less frequently used as quality cues by these consumer groups. It is important to strengthen the institutional framework related to food safety and quality following best practices from EU countries.

## 1. Introduction

Food quality can be defined as “fitness for consumption” and as “the requirements necessary to satisfy the needs and expectations of the consumer” [1] (p. 4). Food quality consists of an objective and a subjective dimension [2], and food producers are only successful if they are able to combine these two dimensions by meeting consumers’ expectations (i.e., subjective quality) and transforming these into specific physical attributes (i.e., objective quality). Subjective food quality often refers to process attributes such as organic production or attention to animal welfare, but it also covers attributes such as taste or price [2]. Objective quality covers the chemical, microbiological, and physical attributes of a food product. For example, the origin of food influences a subjective quality perception by consumers and has an objective quality difference based on the “terroir”, the unique constellation of micro climate, soils, precipitation leading to a unique composition of nutrients in foods of different origin [3].

“Quality cues are defined as information stimuli that are related to the quality of the product and can be ascertained by the consumer through the senses prior to consumption” [4] (p. 312). To form a quality judgement, consumers can draw from single quality cues or a combination of cues [5]. When a desirable product attribute is unknown, consumers infer it from quality cues, which can be any evidence or information that consumers believe may be predictive of the desired attribute [6]. The physical characteristics of the product can also be used as cues: for example, the color or other aspects of the product’s appearance may be used as an indicator of the product quality [7]. The fact that some quality cues can be ascertained prior to consumption means that these quality cues are either search (e.g., brand name, food safety certificates) or credence attributes (e.g., food certificates, organic production); this differentiation between search, experience and credence attributes goes back to the early 1970s [8,9,10]. Search attributes are attributes that can be inspected by the consumer in the store or the supermarket and are used as quality cues to predict the quality of the food product [11].

Food safety, from a consumer’s perspective, is part of the perceived (subjective) quality of a food product and is considered to be an inherent part of food quality [1]. Food safety is a credence attribute, as it is unknown before the purchase whether the food is in fact safe; safety cannot be ascertained by experience after purchase either [2], unless, of course, the consumer becomes sick and the sickness can be attributed to the food with some certainty.

West Balkan Countries (WBCs) belong to the countries with higher corruption and lowest GDP (Gross Domestic Product) in Europe. Insofar it is not surprising that their national food safety control systems are characterized by incoherent legislation and a lack of human and capital resources, as well as reliable data [12], resulting in weak law enforcement. Even though legislation improved food safety regulations during the last years (also due to the growing support and pressure emerging from the EU integration process as well as the request of international food companies requiring certifications of food safety standards) [13], these problems still exist and create real and perceived safety risks for consumers.

Developing and transition countries face serious challenges related to food safety due to weak animal disease controls that result in a higher prevalence of endemic infectious animal diseases [14]. Problems in the agricultural health and food safety systems have been identified by several studies, especially in the meat [15,16] and dairy sector [17,18]. Brucellosis has been a major health concern in Albania and Kosovo [17,19], and aflatoxin in maize for feed was reported in other WBCs [20,21]. Reportedly, aflatoxin also represents a serious problem for small dairy farmers in Albania and Kosovo [22,23], and there are similar concerns in Serbia [18]. In general, mycotoxins and aflatoxins are reported to be of high significance concerning food safety in these transition economies [24]. Other studies report high levels of contamination of plants and soil with cadmium, lead, and other heavy metals in these countries [25].

Public agencies are responsible for ensuring food safety enforcement; however, their capacity in the context of developing or transition countries (like Albania or Kosovo) is limited, conditioned by the weak institutional framework and corruption [26]. This finding is especially relevant because consumers tend to derive food safety based on their trust into regulators and food manufacturers [27]. Due to the lack of capacity of public agencies Albanian consumers trust more in retailers than in governmental regulators for guaranteeing food safety [15]. Due to this context Albanian consumers also prefer to buy food directly from producers [28]. In Albania and Kosovo, most farmers across agri-food sectors lack information or awareness related to food safety standards. More specifically, in the case of the livestock sector (which is also the focus of this paper and also which among most sensitive sectors related to food safety), most farmers lack information about which institutions are in charge of basic standards control related to food safety or animal welfare—lack of awareness about standards results in standards non-compliance [17,29].

Considering these food safety issues, consumers in WBCs unsurprisingly perceive food safety as critical. However, consumers are in general unfamiliar with international food safety standards [15,30]. Zaric et al. [31] found that quality is by far the most important factor influencing consumer purchasing behavior in Serbia. A previous study found that Kosovar consumers perceive origin, food safety certificates, and brands as important means to identify food safety and quality [32]. Brand reputation is a widely used cue to reduce the perceived risks associated with food purchases. Strong brand reputation leads to higher perceived quality [5,33]. Consumers also use provenance of a food product—may it be regional, domestic, or international—as an indicator for quality [34,35,36] but “indication of origin may only become a signal of enhanced quality if the source of origin is associated with higher food safety or quality” [37]. A study by Miftari [38] showed that a majority of Kosovar consumers had a positive bias towards domestic (versus foreign) dairy products, which points towards a prevalence of consumer patriotism concerning food. A qualitative means-end chain analysis of Croatian consumers concerning their motives for buying traditional food products found that domestic origin was an important quality cue that consumers connected with the absence of a risk to personal health. The Croatian consumers even associated a domestic origin with better flavor/taste, and connected the attribute “domestic origin” with the use of traditional ingredients and with trustworthy producers [39]. Food labels and certificates can act as quality cues if they are available, are understood by consumers, and are regarded as trustworthy [2]. Verbeke and Ward [36] found that there is high consumer interest in quality seals and expiration dates as quality cues. Consumers also use organic production (a specific form of a food certificate) as a quality cue. The extent to which they use organic food certificates as a quality cue is influenced by their environmental concerns, their trust in local producers in the country and their ethnocentrism [40]. According to a previous consumer study in Albania, most consumers consider factors surrounding health to be the most important dimension of the organic products, while the impact of organic food production on the environment does not appear to be important at all [41]. Comprehensive information about ingredients and the origin of raw materials is an important cue to create consumer trust in brands, based on a broad study of German consumers [42]. In the absence of trustworthy third-party certification of food products, personal trust between customer and seller can compensate as a quality cue. The reputation or personal knowledge of the seller or manufacturer of a food product is used to judge the quality and safety of food products [43,44]. For example, the majority of respondents of a study analyzing consumer behavior in Albania (purchasing lamb meat) answered that their main source of trust is knowing the butcher or seller [16].

Rising consumer concern about food safety has led to an increased demand for standards related to quality assurance such as ISO 9000 or HACCP, which focus on objective food quality [45,46]. Despite the fact that HACCP and ISO 9000 have been implemented in the market for decades, most consumers do not know about them [47]. This is not surprising, because they have been established as business-to-business standards, often treated by retailers as an insurance policy to protect them against food scandals [48]. WBCs are known for weak governmental institutions. This is the reason why consumers in Albania trust more the retailer concerning food safety [15] and if the option is available, consumers tend to buy directly from the farmer, because they believe the food to be of higher quality and safer [28]. For example, the biggest knowledge gaps among food handlers in Serbia are related to temperature control and sources of food contamination [49]. In response to a possible EU accession in the future, WBCs are in the process of implementing and harmonizing their food quality and safety standards, and it is also necessary for them to fulfill these standards for trade with the EU.

Perception of food safety might also be affected by socio-demographic factors such as gender and education. In particular, female and better educated persons are assumed to pay more attention to food safety and related issues [16,50]. Gkana and Nychas [51], in a study of Greek consumers, found no differences in gender and age in respect to perceptions of food safety, but a higher education level resulted in a higher awareness and knowledge of food safety issues. A study of Turkish consumers reported that households with higher income and higher education levels were more interested in food safety, and so were female and older consumers [52]. Another study of Turkish consumers found a positive influence of education on knowledge of, attitudes towards, and practices related to food safety [53].

In a nutshell, previous studies looked at singular aspects, how consumers in WBCs evaluate food quality and safety. To our knowledge, none of the previous studies collected data in more than one country. This study is the first one to deliver comparative results for Albania and Kosovo. The consumer survey is based on the evaluation of milk and cheese products, for which food safety is very sensitive. Accordingly, milk and cheese products serve as a representative food product category to measure consumers’ perception of quality and safety standards.

## 2. Materials and Methods

A quantitative, structured survey with urban consumers from Albania and Kosovo was conducted in 2019, with the data being collected in Prishtina and Tirana. These are the capital and largest cities of the respective countries (home to almost 1/3 of the population), where purchasing power is concentrated, and as such, the findings are important for the industry. Furthermore, both cities have high diversity of cultures and origins (most residents have migrated from all parts of the respective countries, including rural areas), and as such, the findings can be considered indicative also for the rest of the country. In April and May 2019 students, previously trained in taking face to face interviews, collected the data in Prishtina (Kosovo) and Tirana (Albania). The interviews were done outdoor on market places and public squares by using a convenience sample technique without quotas.

Within the questionnaire, two important batteries of items measured the perception of food safety and quality; the first had a 5-point Likert scale for the perception of food safety and quality for domestic milk and cheese, taken from the literature [54], while the second battery of items presented a list of quality and safety attributes. These attributes were taken from the literature review above and included brand reputation, expiry date, list of ingredients, organic production, food safety certificates, HACCP and ISO 9000, origin (local, domestic, foreign, or EU), and, finally, knowledge about the seller and the producer. We asked the consumers how often they bought cheese and how often they used these attributes related to quality and food safety. The answers measured the frequency, from 1 = never to 5 = always. Pre-test interviews to check the questionnaire were conducted in Pristina and Tirana with randomly selected consumers. The intended sample size was 300 interviews for each country. Based on the literature review, we formulated three hypotheses for further testing:The quality and safety of imported food is rated higher than the quality and safety of domestic food. In both countries, the import of cheese is dominated by EU countries where the food safety standards tend to be much higher (imports from non-EU countries are negligible).Kosovo (Prishtina) and Albania (Tirana) differ with respect to the frequency of use of cues related to food safety and quality.Socio-demographic variables influence the perception of food safety and quality cues.

We used the *t*-test to identify differences between populations, and correlation analysis. In order to improve the reliability of the test results, we implemented the bootstrapping method [55]. Bootstrapping is usually used when a standard normal distribution cannot be guaranteed. It is “a computationally intensive method that involves repeatedly sampling from the data set and estimating the indirect effect in each resampled data set” [55] (p. 80). Developed in the 1970s [56], the bootstrapping method supports the reliability of analytical interpretations in quantitative research [57] and delivers valuable information about a possible distribution of analytical findings. There are a number of comparable publications available that incorporate bootstrapping into quantitative analysis to improve reliability to test, for example, the significance of differences in consumer evaluations between populations [58]. In accordance with recently published research about consumer perception and food safety, we used 5000 bootstrap samples and the 95% bootstrap confidence interval [59].

To test the hypotheses, we collected data in Prishtina (*n* = 300) and Tirana (*n* = 349). The overall sample size therefore amounts to 647 valid responses. The samples had a similar structure with respect to age and household size to statistical data for the populations of Kosovo and Albania. The samples were, however, biased with respect to the variables “gender” (more females than males in the Albanian sample), “education level” (the samples were more highly educated) and, to some extent, “income”—the structure of the income across the sample was not completely comparable to the general statistics, since the distribution of the income within the sample seems to deviate towards a higher income compared to the overall population (Table 1). Therefore, the following analysis will probably not deliver perfectly transferable results. Although this is not the main intention of this study, we have to take this into account when interpreting the following results.

## 3. Results

Confirming the literature review, there are obviously certain problems with food standards in Kosovo and Albania. Therefore, we assumed that the safety and quality of imported food would be rated higher than the safety and quality of domestic food (tested on the example of cheese and fresh milk). To assess the perception of domestic food in comparison to imported food, the respondents were asked to specify their agreement with four statements concerning the safety and quality of domestic and imported foods, by means of a 5-point Likert scale (1 = total disagreement with the statement; 5 = total agreement with the statement). In addition, the respondents were asked whether they considered imported food to be of high quality.

The descriptive analysis clearly shows that in Kosovo (Prishtina), consumers considered domestic dairy food to be much safer and of much higher quality than imported dairy food (column *M* “Prishtina” in Table 2). Also, in Albania (column *M* “Tirana” in Table 2), the four comparison statements (domestic vs. imported dairy food) on average received an agreement score of more than three (i.e., a greater preference for domestic dairy food). However, the values are much lower than for the Kosovo sample and much closer to the mid-value of three (3 = “neither agree, nor disagree with statement”, which should be interpreted as a perception of no difference between domestic and imported dairy food). For both samples, the average for the last “imported cheese” statement was around three: no clear tendency is visible and we assume that, in general, imported cheese is not considered to be of a high quality.

To test whether the mean for the perception of domestic vs. imported food (*M*) differs significantly between the Prishtina and the Tirana samples, we used an unpaired two sample *t*-test including bootstrap sampling with 5000 random bootstrap samples (Table 2). The differences for the four comparison statements were significant (*p* ≤ 0.001), and most of them were considerable (*MD* = +0.814 to +1.080; *t* = 8.6 to 12.8). The 95% confidence interval based on bootstrap sampling confirms this interpretation: the conclusion is that Prishtina consumers have significantly more trust in domestic products. The value for *M* is considerably higher, which means that these consumers agreed more strongly with the four comparison statements and rated domestic products as safer and of higher quality than imported products.

The following test of H1 shows whether the deviations from three are significant. Based on the descriptive results and the results of the *t*-test presented above, we assume that the Kosovo sample has, in general, a much higher preference for domestic food.

### 3.1. Test of Hypothesis 1: Perception of Food Quality and Safety of Imported and Domestic Food

As a result of certain food safety problems in the WBCs, we formulated H1: The quality and safety of imported food is rated higher than the quality and safety of domestic food. However, the descriptive results clearly suggest that H1 should be rejected and that, in general, domestic food is rated as safer and of higher quality. Obviously, consumer patriotism is relevant for Kosovar, and also Albanian, consumers. The following analysis tests whether the responses to the five statements in Table 3 are significantly higher or lower than three (“neither agree, nor disagree with statement”). If the differences are significant, we interpret the deviations in view of their metric size (positive or negative). If they are not significant, there is no empirical evidence that imported and domestic food were rated differently, and, instead, domestic and imported food were considered to be equal in view of safety and quality. To analyze the deviations, we performed a *t*-test with a test value of three including a bootstrapping method with 5000 random bootstrap samples.

The means *M* of the Prishtina and Tirana samples are found to be significantly higher/lower than three. The *t*-test with bootstrapping shows that the deviations within the Prishtina sample (*n* = 300) are highly significant for the comparison statements (*M* = 4.13 to 4.36; *t* = 19.5 to 26.1; *p* ≤ 0.001; Table 3). All the deviations are significantly positive (*MD* = +1.133 to +1.363), meaning that the respondents on average clearly agreed with the statements that domestic food is safer and of higher quality than imported food. The deviations from three for the “imported cheese” statement are not significant (*M* = 2.93; *MD* = −0.067; *t* = −1.095; *p =* 0.275; Table 3). Within the Tirana sample (*n* = 349), the deviations are also significant (except for the “imported cheese” statement), but they are much lower (*M* = 3.23 to 3.33; *t* = 3.448 to 4.772; *p* ≤ 0.001; Table 3). Again, with respect to the “imported cheese” statement, the deviations from three are not significant (*M* = 2.96; *MD* = −0.043; *t* = −0.776; *p* = 0.438). The 95% confidence intervals of the bootstrapping show the principal reliability of these results and that it is advisable to reject H1: The quality and safety of imported food is not rated higher than the quality and safety of domestic food. In Prishtina, consumers generally evaluated domestic food to be safer and of higher quality. In Tirana, too, the consumers tended to evaluate domestic food more highly (there are significant, positive deviations from three); however, although the deviations are significant, they are, on average, much lower, and the results are therefore much less clear than for the Prishtina sample.

### 3.2. Test of Hypothesis 2: Perception of Food Safety and Quality Cues

To analyze perceptions of food safety and quality cues, the respondents were asked how often they checked specific characteristics that are connected to food safety and quality (tested on the example of cheese). The relevant question (Q11) was: “If you want to know about the safety of cheese you are going to buy, which of the following characteristics do you check?” The respondents used a 5-point semantic scale, with a minimum value of 1 (never) to a maximum of 5 (always). The semantic meaning of the in-between values are: 2 = occasionally (about 1 to 2 times per week), 3 = frequently (about half the time or 3 to 4 times every week); 4 = often (about 5 times per week).

As we can see from Table 4 the consumers in Prishtina checked the food safety and quality cues much less frequently than those in Tirana—with one exception, the expiration date. Overall, the expiration date was the most important food safety and quality cue for consumers in both samples. Other important cues were the list of ingredients, organic production, origin, and knowledge of the producer and seller (particularly for consumers in Tirana). The food safety and quality cues with the lowest importance (within both samples) were the international food standards HACCP and ISO. To see whether these differences in the perception of food safety and quality cues are significant, we tested Hypothesis 2.

According to Hypothesis 2 consumers from Kosovo (Prishtina) and Albania (Tirana) should differ in respect to the frequency at which they used food safety and quality related cues. This hypothesis was proposed because the average income in Albania is much higher than the average in Kosovo, and the level of education is also higher in Albania (see sample description in Table 1). As we can see from Table 4, most quality cues were rated to be more important (that is, the consumers checked them more regularly) by consumers in Tirana. The mean differences are quite large. Neither of the samples identified safety and quality cues that were of the highest importance, with average values greater than four, and it was only “expiration date” that reached a mean close to four. In particular, in Prishtina only a few food safety and quality cues seem to have high importance. We tested the differences between the samples by means of an unpaired two sample *t*-test including bootstrap sampling with 5000 random bootstrap samples (Table 4). All of the differences were significant, and all, except for the differences for expiration date, were considerable. *MD* ranges from −0.148 (expiration date) to −1.236 (organic production). This means that consumers from Tirana seem to use food safety and quality cues much more frequently than consumers from Prishtina. Bootstrapping further emphasizes the accuracy of this interpretation, with the 95% confidence intervals shown in Table 4.

### 3.3. Test of Hypothesis 3: The Influence of Socio-Demographic Variables on the Perception of Food Safety and Quality Cues

As pointed out in the literature, socio-demographic factors might influence the perception of food and quality cues, which leads to H3: Socio-demographic variables influence the perception of food safety and quality cues. In particular, we tested the variables “gender”, “age”, “household size”, “education”, and “income” as predictors for the rating of domestic in comparison to imported food and the perception of food safety and quality cues. To evaluate the significance of the socio-demographic variables we (1) compared means by the use of appropriate statistical methods (independent *t*-test, correlation analysis); and (2) if significant differences were found, analyzed the effects of the independent variables as predictors in order to ensure the reliability of the analysis, including the use of bootstrapping (5000 bootstrap samples; valid data without missing values in Prishtina *n* = 299 and in Tirana *n* = 344).

As we can see from Table 5 (which shows only the significant correlation coefficients), the dependency of the perception of domestic in comparison to imported dairy food on socio-demographic variables is rather low or almost non-existent. Significant correlation coefficients were only found for education and household size (in Prishtina), and age and, to some extent, education in Tirana. The variables “gender” and “income” had no influence in either sample. Concerning statements one to four (domestic vs. imported foods), in Prishtina, there was a weak negative correlation to education and a weak positive correlation to household size. This could be interpreted as showing that well educated people had (slightly) more trust in imported food than less well-educated people, and that larger households perceived domestic dairy food as more trustworthy. In contrast to these findings for Prishtina, the Tirana respondents showed different patterns, where age seems to have an influence, but a very small one, on the perception of domestic versus imported dairy food, and education seems to have a positive influence on the statement that imported cheese is of higher quality. However, even the significant correlations are quite low, with the minimum and maximum *r* being −0.222 and +0.299, respectively. Given the bootstrap 95% confidence interval, *r* would not undercut −0.325 or exceed +0.391. Even the minimum/maximum correlation coefficients do not reach a moderate *r*-level of beyond or below ±0.5.

Concerning the perception of food safety and quality cues, the influence of socio-demographic variables seems to be more important than in the previous findings. In particular, the gender of the respondents seems to have a significant influence on the perception of food safety and quality cues (Table A1 in the Appendix A). In general, women considered food safety and quality cues, such as lists of ingredients or seals of quality, more often than the men (negative *MD* means that the average Likert value for the men is lower than that for the women; the men less often checked food safety and quality cues than the women). Table A1 in the Appendix A contains only significant differences for both samples. As we can see from A1, the findings are quite similar, although some differences can be observed: origin seems to be more relevant for gender differences in the Kosovar sample, while brand reputation and certificates (HACCP, ISO 9000) seem to be slightly more important in the Albanian sample. After bootstrap sampling, the 95% confidence interval shows, however, that the “true” difference in the perception between men and women might—although it is significant—be much lower or higher. It appears that for some variables the differences might also be almost non-existent (the lower boundary of the bootstrap 95% confidence interval is close to 0; e.g., for local origin). For other variables (e.g., organic production), these differences are well supported by the bootstrapping method.

Concerning the other socio-demographic variables, education, in particular, seems to influence the perception of food safety and quality cues significantly in both samples. In the Kosovar sample, income slightly influences the perception of food safety and quality cues (in particular, brand reputation and knowledge of producer). Further, the negative *r* for the variable “household size” might be negligible (Table A2 in Appendix A). In the Albanian sample, age also seems to have a negative, but minor, influence on the perception of the food safety and quality cues (the older people less frequently checked the cues). However, here too, the influence approximated through correlation analysis is rather low for almost all the socio-demographic variables (see *r* in Table A2 for the Kosovar sample and in Table A3 for the Albanian sample). These findings are further supported by the bootstrapping method; no upper/lower boundaries in the 95% confidence interval exceed/undercut ±0.5—a level at which one could suppose at least a moderate effect of socio-demographic variables on the perception of food safety and quality cues.

In a nutshell, one might consider socio-demographic variables as relevant predictor variables. Even though their influence seems to be rather low, H3 is clearly supported: Socio-demographic variables do have an influence on the perception of food safety and quality in Kosovo and Albania.

## 4. Discussion

Based on the literature review [15,16,17,18,32], which indicated a lower level of food safety in Albania and Kosovo [22,23] (and other WBCs [20,25]), we expected that respondents from both countries would evaluate imported better than nationally produced food products (H1). However, we did not observe this in our study, so H1 could not be confirmed and had to be rejected. The descriptive analysis and the test for significant differences showed that in Kosovo (Prishtina) consumers evaluated domestic dairy food to be safer and of higher quality than imported dairy food, which is in accordance with [38]. Albanian (Tirana) consumers also showed a greater preference for domestic over imported cheese and milk products. However, the values are lower than in the Kosovar sample. Both samples were indifferent about the quality of imported cheese. Consumer patriotism also seems to be of relevance for Kosovar and Albanian consumers.

For H2, we assumed that Kosovar consumers would differ from Albanians in respect to the frequency at which they used food safety and quality related cues. We used a list of quality and food safety cues derived from the literature to test this assumption. Important quality and food safety cues mentioned in the literature are brand reputation [5,33], information on labels (expiry date, list of ingredients) [36], food certificates [36], organic production [2,40], quality related standards (ISO 9000, HACCP) [45,46], and country of origin [32,34,35,36,39]. In the absence of reliable quality standards or third party food certificates, knowing the producer or seller of a product can act as another quality cue [44]. H2 was confirmed, with the differences between Kosovar and Albanian consumers being significant for all the food safety and quality cues tested.

In our study, the Kosovar consumers used food safety and quality cues less frequently than Albanians. Expiration date was the most commonly used quality cue for both consumer groups. The Albanians used food certificates more often than the Kosovars. HACCP or ISO related information was the least frequently used quality cue for both consumer groups, which is not surprising because HACCP and ISO standards are primarily used for business-to-business communication and are normally not communicated to consumers. The frequency of use (see Table 4) showed that information about expiry date, domestic origin/local origin, organic production, knowing the producer, and brand reputation were the most frequently used food safety and quality cues for the Albanians. The Kosovar consumers used expiry date, domestic origin, knowing the producer, brand reputation and local origin, in descending frequency. The Kosovar respondents used information about organic production and food certificates much less often than the Albanian respondents (see Table 4, biggest mean difference of 1.236 (organic) and −1.201 (certificates)). This may be an indicator of a lower availability of organic food supplies in Kosovar or also can reflect the strategy of Albanian consumers to pay more attention to organic certification to tackle their concern for food safety situation [41]. As suggested by the study of Thøgersen et al. (2019) it could also be an indicator of an absence of environmental concerns among consumers or insufficient trust in the countries of origin of the organic food [40]. We found that food certificates were less frequently used by Kosovars than by Albanians. According to Grunert (2005), this could be an indicator either that the Kosovar consumers did not consider the food certificates on dairy products to be trustworthy, because of the lack of trust in formal institutions [26] or that they did not understand them [2]. In our sample, Kosovar consumers used the quality cue “knowing the producer” in third place after expiry date and domestic origin, which is in accordance with several international studies and studies from the WBCs [15,28,43,44]. In the absence of trustworthy third-party certification of food products, the personal trust between customer and seller can compensate as a quality cue.

Previous studies have reported that female consumers and better educated consumers pay more attention to information related to food safety [16,50]. Our study found similar, statistically significant results (H3: Socio-demographic variables influence the perception of food safety and quality cues). Women and better educated consumers checked food safety and quality related information more often than men or those with lower education. Women obviously considered food safety and quality cues, such as list of ingredients or food certificates, more often than men. Origin seemed to be more relevant with respect to gender differences in the Kosovar sample, while brand reputation and quality assurance standards (HACCP, ISO 9000) seemed to be slightly more important in the Albanian sample. The other socio-demographic variables (age, income, household size) were either not statistically significant or, if they were significant, showed only small correlation coefficients. Older Albanian consumer groups, for instance, perceived the quality and safety of domestic dairy products to be higher than did younger consumer groups, which makes sense as older consumer groups tend to show more ethnocentrism than younger consumer groups (see [53]).

## 5. Limitations and Conclusions

The major limitation of this study, besides the convenient nature of the samples, is the fact that the respondents were selected from the countries’ two main cities, and thus the samples do not represent all consumers but rather urban consumers. As such, the study findings cannot be considered representative of the whole population. However, as highlighted earlier in the paper, Tirana and Pristina are the capital and largest cities of the respective countries, with high diversity of cultures and origins, and as such, the findings can be considered indicative also for the rest of the country. Furthermore, these two cities also represent the largest markets where purchasing power is concentrated, and as such, the findings are important for the industry. Future studies could broaden the context of this study by including social and cultural conditions that are perhaps able to further explain differences in consumer perspectives between Kosovars and Albanians on food safety and quality.

Confirming literature [12], in Albania and Kosovo there are serious gaps in the food safety system (in particular in the meat [15,16] and dairy sector [17,18]), which is reflected in real and perceived food safety risk. However, according to the study findings, overall, consumers have higher preference for local products, the safety and quality of which is perceived higher compared to imports, although cheese imports are dominated by EU countries, which are characterized by high food safety standards. Kosovar consumers’ perception of food safety is higher when compared to Albanians—one factor that may contribute to this difference is the fact that Kosovo has been under the supervision and heavily supported by the international community in its efforts to build up the institutional system after the war. Another reason could be a high prevalence of ethnocentrism of Kosovar consumers, as identified in a previous study [38].

As expected, women and higher educated consumers used food quality and safety related information more often than other consumer groups in both countries. In accordance with this result of our study, it is advisable for governmental institutions who want to inform the public or food companies releasing advertising campaigns to focus on these consumer groups to improve the effectiveness of their communication efforts. However, stakeholders could also focus especially on less well-educated consumers to reduce their information deficits concerning food safety and quality standards in their home countries.

We found that Kosovar consumers used food certificates less often than Albanians. This could be either because they do not trust them or because they do not understand them. In the first case, this would emphasize the need to establish trustworthy third-party certification, while the latter case would point towards a promotion of existing food certificates by governmental or privately funded educational campaigns (see [53]). In our opinion, these findings could be used by policymakers and food companies to improve the level of trust in public/private institutions to guarantee food safety, and to increase consumers’ awareness and knowledge in this area. They underline the importance of food labeling information such as expiry date/best before date, of trustworthy food brands and of clear and transparent communication of the country of origin or the local origin.

Although not directly addressed within this study, we have to take into account that there is still a lack of capacity due to weak institutional frameworks and corruption [26]—despite the improved legislation in Kosovo and Albania in recent years due to EU integration processes [13]. An important institutional measure could be the strengthening of the existing or establishment of new national food safety and quality organizations, following best practices from EU countries, such as AMA Marketing in Austria (AMA = Agricultural Market Austria). AMA Marketing is a governmental organization independent from existing ministries, comparable to SOPEXA in France, with a focus on establishing and controlling food quality and safety standards, and implementing communication strategies to reach consumers. The advantage of establishing a new organization independent from existing agricultural or health ministries could be an important sign to consumers that the new organization also stands for an improved, trustworthy regulatory capacity. This approach could help to gain more trust in the national food safety system and help to further improve the food safety perception of consumers, which is advisable, according to our empirical findings.

## Figures and Tables

**Table 1 foods-10-00160-t001:** Socio-demographic variables of the sample and population.

		Sample % *n* = 300	Kosovo % ^1^	Sample % *n* = 349	Albania % ^2^
Age	15–24	9.4	15.0	11.5	11.0
	25–40	34.2	36.8	34.9	27.9
	41–54	35.9	26.3	25.4	24.6
	55–64	14.4	11.0	15.3	17.9
	65+	6.0	10.8	13.0	18.7
Gender	Male	42.0	50.3	35.5	49.9
	Female	58.0	49.7	64.5	50.1
Education	Basic to Middle School	24.0	66.5	10.5	58.8
	High School	30.3	20.6	35.2	29.9
	Higher Education, University	45.7	12.9	54.4	11.3
Household size	1–2 persons	4.7	9.3	22.1	28.2
	3 persons	7.7	8.3	16.9	19.5
	4 persons	22.3	15.8	29.4	26.8
	5 persons or more	65.3	66.6	31.7	25.5
Monthly income	Up to 500 EUR	14.0		42.7	
	501 to 800 EUR	32.7		41.9	
	801 to 1200 EUR	35.7		12.2	
	More than 1200 EUR	17.7		3.2	
	Mean (approx.) per capita		370		330

^1^ Kosovo Agency of Statistics, 2017 (http://askdata.rks-gov.net); ^2^ Institute of Statistics for Albania (http://www.instat.gov.al/): Age, gender (total population): 2019; education: Census 2011, household budget survey 2018.

**Table 2 foods-10-00160-t002:** Differences in the assessment of domestic in comparison to imported dairy food in Kosovo (Prishtina) and Albania (Tirana) (*n* = 649); unpaired two sample *t*-test including bootstrap sampling.

	*M*		*MD*	*SE*	*t*	*p*	Bootstrap ^3^		
Statements ^1^	Prishtina	Tirana					*MD* 95% Confidence Interval		*p*
Domestic cheese is safer than imported ^2^	4.14	3.33	+0.814	0.095	8.636	≤0.001	+0.628	+1.001	≤0.001
Domestic milk is safer than imported ^2^	4.19	3.29	+0.894	0.096	9.474	≤0.001	+0.704	+1.077	≤0.001
Domestic cheese is of higher quality than imported ^2^	4.36	3.28	+1.080	0.086	12.842	≤0.001	+0.913	+1.245	≤0.001
Domestic milk is of higher quality than imported ^2^	4.13	3.23	+0.901	0.090	10.151	≤0.001	+0.723	+1.072	≤0.001
Imported cheese is of high quality	2.93	2.96	−0.024	0.082	−0.288	0.773	−0.182	+0.144	0.772

*M* = Mean; *MD* = Mean Difference; *SE* = Standard Error; *p* = Significance; ^1^ Likert scale: 1 = “disagree” to 5 = “agree”; ^2^ Levene test on homogeneity of variances: variances are not equal, Welch test used; ^3^ 5000 bootstrap samples.

**Table 3 foods-10-00160-t003:** Perception of domestic in comparison to imported dairy food in Kosovo (Prishtina) and Albania (Tirana); *t*-test including bootstrap sampling with test value = 3 (“neither agree nor disagree with statement”).

Statement ^1^	*M*	*MD*	*SE*	*t*	*p*	Bootstrap ^2^		
						*MD* 95% Confidence Interval		*p*
Kosovo, Prishtina (*n* = 300)								
Domestic cheese is safer than imported	4.14	+1.143	0.065	17.830	≤0.001	+1.014	+1.272	≤0.001
Domestic milk is safer than imported	4.19	+1.187	0.064	18.640	≤0.001	+1.062	+1.314	≤0.001
Domestic cheese is of higher quality than imported	4.36	+1.363	0.053	26.172	≤0.001	+1.260	+1.466	≤0.001
Domestic milk is of higher quality than imported	4.13	+1.133	0.058	19.575	≤0.001	+1.021	+1.244	≤0.001
Imported cheese is of high quality	2.93	−0.067	0.061	−1.095	0.275	−0.184	+0.055	0.275
Albania, Tirana (*n* = 349)								
Domestic cheese is safer than imported	3.33	+0.330	0.069	4.772	≤0.001	+0.194	+0.463	≤0.001
Domestic milk is safer than imported	3.29	+0.292	0.069	4.193	≤0.001	+0.156	+0.426	≤0.001
Domestic cheese is of higher quality than imported	3.28	+0.284	0.066	4.299	≤0.001	+0.156	+0.411	≤0.001
Domestic milk is of higher quality than imported	3.23	+0.232	0.067	3.448	0.001	+0.099	+0.362	0.001
Imported cheese is of high quality	2.96	−0.043	0.055	−0.776	0.438	−0.152	+0.064	0.433

*M* = Mean; *MD* = Mean Difference; *SE* = Standard Error; *p* = Significance level; ^1^ Likert scale: 1 = “disagree” to 5 = “agree”; n = 300; 299 degrees of freedom (*df*) (Prishtina Sample); n = 349; 348 *df*; (Tirana Sample) ^2^ 5000 bootstrap samples.

**Table 4 foods-10-00160-t004:** Differences in the rating of food safety and quality cues between consumers from Kosovo (Prishtina) and consumers from Albania (Tirana) (*n* = 649); unpaired two sample *t*-test including bootstrap sampling.

	*M*		*MD*	*SE*	*t*	*p*	Bootstrap ^3^		
Characteristics ^1^	Prishtina	Tirana					*MD* 95% Confidence Interval		*p*
Brand reputation	2.41	3.13	−0.723	0.076	−9.499	≤0.001	−0.869	−0.576	≤0.001
Expiration date ^2^	3.63	3.78	−0.148	0.051	−2.900	0.004	−0.249	−0.045	0.003
List of ingredients ^2^	2.54	2.86	−0.322	0.116	−2.787	0.005	−0.550	−0.097	0.004
Organic production ^2^	1.94	3.17	−1.236	0.104	−11.909	≤0.001	−1.440	−1.027	≤0.001
Food safety certificate	1.86	3.06	−1.201	0.092	−13.009	≤0.001	−1.377	−1.019	≤0.001
HACCP ^2^	1.35	2.11	−0.756	0.077	−9.832	≤0.001	−0.907	−0.611	≤0.001
ISO ^2^	1.36	2.14	−0.782	0.078	−10.038	≤0.001	−0.935	−0.633	≤0.001
Local origin ^2^	2.25	3.24	−0.983	0.085	−11.515	≤0.001	−1.149	−0.813	≤0.001
Domestic origin	2.87	3.24	−0.360	0.078	−4.639	≤0.001	−0.511	−0.214	≤0.001
Foreign origin	2.16	2.93	−0.777	0.085	−9.183	≤0.001	−0.943	−0.616	≤0.001
EU origin	1.88	2.86	−0.985	0.084	−11.785	≤0.001	−1.143	−0.826	≤0.001
Knowing the seller ^2^	2.10	2.84	−0.742	0.091	−8.147	≤0.001	−0.923	−0.561	≤0.001
Knowing the producer	2.63	3.05	−0.419	0.091	−4.581	≤0.001	−0.594	−0.244	≤0.001

*M* = Mean; *MD* = Mean Difference; *SE* = Standard Error; *p* = Significance; ^1^ Likert scale: 1 = “never check characteristic to assess food safety and quality”, 5 = “always check characteristic to assess food safety and quality”; ^2^ Levene test on homogeneity of variances: variances are not equal, Welch test used; ^3^ 5000 bootstrap samples.

**Table 5 foods-10-00160-t005:** Correlation between socio-demographic variables and perception of domestic in comparison to imported dairy food for Kosovo and Albania; correlation coefficients including bootstrap sampling.

Statements ^1^	*r*	*SE*	*p*	Bootstrap ^2^ *r* 95% Confidence Interval	
Prishtina: Education	Spearman-Rho				
Domestic cheese is safer than imported	−0.222	0.053	≤0.001	−0.325	−0.118
Domestic milk is safer than imported	−0.174	0.054	0.003	−0.281	−0.065
Domestic cheese is of higher quality than imported	−0.147	0.056	0.011	−0.256	−0.038
Domestic milk is of higher quality than imported	−0.143	0.056	0.013	−0.254	−0.033
Prishtina: Household size	Pearson’s *r*				
Domestic cheese is safer than imported	0.287	0.049	≤0.001	0.190	0.383
Domestic milk is safer than imported	0.299	0.048	≤0.001	0.205	0.391
Domestic cheese is of higher quality than imported	0.226	0.054	≤0.001	0.119	0.330
Domestic milk is of higher quality than imported	0.194	0.056	0.001	0.084	0.301
Tirana: Age	Pearson’s *r*				
Domestic cheese is safer than imported	0.185	0.053	0.001	0.080	0.286
Domestic milk is safer than imported	0.139	0.052	0.010	0.035	0.239
Domestic cheese is of higher quality than imported	0.135	0.055	0.012	0.026	0.241
Domestic milk is of higher quality than imported	0.108	0.056	0.045	−0.003	0.216
Imported cheese is of high quality	−0.192	0.054	≤0.001	−0.299	−0.081
Tirana: Education	Spearman–Rho				
Imported cheese is of high quality	0.210	0.053	≤0.001	0.104	0.311

*r* = Correlation coefficient Spearman-Rho (ordinal data) or Pearson’s *r* (metric data); *SE* = Standard Error; *p* = Significance; ^1^ Likert scale: 1 = “disagree” to 5 = “agree”; ^2^ 5000 bootstrap samples.

## Data Availability

The data presented in this study are available on request from the corresponding author. The data are not publicly available due to privacy.

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
