# Peer review of "Consumer Perception of Food Quality and Safety in Western Balkan Countries: Evidence from Albania and Kosovo"

_foods, 2021, doi:10.3390/foods10010160_

Round 1

Reviewer 1 Report

  • Introduction: it is well written and clearly show readers why the research topic is worth reading about and why the paper warrants their attention.
  • Materials and methods: the authors created their questionnaire based on the literature. They also tested their questionnaire in conditions as similar as possible to the research. Statistical tests are described as fully as possible. Besides, in this section the authors also describe the limitations of their study (i.e. prevalence of women).
  • Results: this section is logically arranged and clearly written.
  • Discussion and conclusion: the authors tie properly the different findings together, analyse them in the context of existing literature and, in the conclusion section, highlight the study’s significance and possible impact. 

Author Response

Dear Reviewer1,

thank you for your comments, we really appreciate your time and efforts. We are very glad you liked our manuscript.

Reviewer 2 Report

The goal and content are consistent with the title of the paper. The structure of the paper is correct. The methods were chosen correctly. The objection may concern the relatively small size of the sample and its structure, but the authors are aware of this fact and take it into account when analyzing and interpreting the results. However, the paper requires some adjustments:

  1. In the "Abstract", one or two sentences describing the research methods used should be added. Conclusions resulting from the study should be consistent with those presented in the chapter "Conclusions and limitations".
  2. In the analytical part, there is no information on the diversification of the analyzed problem perception between the respondents living in the countryside and in the cities (the respondents lived mainly in the largest cities). Usually, rural residents trust more domestic and local food, especially staple food products (e.g. bread, milk, cheese). Was it also the case in this study?
  3. The "Discussion" chapter is too short and is limited to presenting the observations resulting from the calculations, without taking into account the broader context, including social and cultural conditions (e.g. based on literature). These "soft" factors would help to explain the observed differences and make the analysis more complete.
  4. Conclusions and limitations are too general in nature and a bit inconsistent. First, it is necessary to synthetically present conclusions, and then make recommendations based on them. In the "Abstract", the authors point to differences in the perception of products between Albania and Kosovo, while in "Conclusions and limitations" differences in education are emphasized. This should be sorted out and possible recommendations should be adapted to at least these two aspects.

Author Response

Dear Reviewer 2, thank you for your time and efforts, please see attachment for our reply.

Reviewer 3 Report

In my opinion the work is well written and discussed with proper references. I suggest minor revisions as follows:  

Line 60: make explicit “GDP”;

Table 1: in the caption, clarify that it is a comparison with the overall population data, as reported in the text;

Table 2 and relative discussion of the results: why the authors performed a paired t-test? As far as I know, paired t-test should be used only when there is dependency between the samples or, in other words, when the observations are related. In the authors work, there should not be any kind of dependence considering that people from Albania and Kosovo are totally different.

The same question also arise considering Table 4.

Lines 304-305: what does means “valid data”? Explain better.

Lines 329-330 and 416-419: it could be that women consider food safety and quality cues more than man because of the frequency of purchase (i.e. purchases are more often made by women then man)?

Line 382: Figure 2?

Author Response

Dear Reviewer 3, thank you for your time and efforts, please see attachment for our reply.

Reviewer 4 Report

The paper presents research into consumer preferences regarding food quality and safety in Kosovo and Albania. This is a very interesting research, but it has one drawback - it concerns two selected regions of southern Europe. The literature review also focuses on the Turkish, Croatian and Greek markets. Which seems too narrow a view. There is also a missing paragraph about the quality assessment methods, e.g. "Application of an electronic nose with novel method for generation of smellprints for testing the suitability for consumption of wheat bread during 4-day storage" or "Aroma effects on food choice task behavior and brain responses to bakery food products cues "etc ... The authors should mention the influence of food origin on its composition, properties, etc. (e.g. "Detection and differentiation of volatile compound profiles in roasted Coffee Arabica beans from different countries using an electronic nose and GC-MS").

"higher educated consumers used food quality and safety related information more often than other consumer groups in both countries" - this is an obvious and expected conclusion.

Besides, the work is well written, statistical methods are used and written in understandable language.

Author Response

Dear Reviewer 4, thank you for your time and efforts, please see attachment for our reply.

Round 2

Reviewer 2 Report

The authors took into account the reviewer’s suggestions. However, the summary should be reorganized to improve its readability. At the beginning of this part of the article there should be an information about the limitations of the study, then the results of observations, and finally recommendations. It should also be clearly stated what is the result of the study and what is the authors’ own opinion. This is necessary, however, the next stage of the review process is not required.

Author Response

Dear Reviewer 2,

thank you once again for your support. According to your comments, we rearranged Chapter 6:

  • we changed the title to "6. Limitations and conclusions" and start it with the limitations of the study.
  • we added some information, where the arguments are coming from: "Confirming literature..." (incl. refs), "according to this result", "in our opinion", inclusion of literature refs
  • the paragraphs in the summary section now clearly distinguish between different issues.
  • we refrained from further separating results and conclusions because we think that the conclusions based on empirical results should not be separated from the relevant empirical results to be traceable.

We guess that these modifications increased the readability of the summary section and are very grateful for your referring comments. Please find the modifications in the attached document.

Kind regards, the Authors
